# Dephosphorylation of the NPR2 guanylyl cyclase contributes to inhibition of bone growth by fibroblast growth factor

Leia C Shuhaibar[1]*, Jerid W Robinson[2†], Giulia Vigone[1†], Ninna P Shuhaibar[1], Jeremy R Egbert[1], Valentina Baena[1], Tracy F Uliasz[1], Deborah Kaback[1], Siu-Pok Yee[1], Robert Feil[3], Melanie C Fisher[4], Caroline N Dealy[4], Lincoln R Potter[2]*, Laurinda A Jaffe[1]*

[1]Department of Cell Biology, University of Connecticut Health Center, Farmington, United States; [2]Department of Biochemistry, Molecular Biology, and Biophysics, University of Minnesota, Minneapolis, United States; [3]Interfakultäres Institut für Biochemie, University of Tübingen, Tübingen, Germany; [4]Center for Regenerative Medicine and Skeletal Development, University of Connecticut Health Center, Farmington, United States

*For correspondence:
shuhaibar@uchc.edu (LCS);
potter@umn.edu (LRP);
ljaffe@uchc.edu (LAJ)

†These authors contributed equally to this work

Competing interests: The authors declare that no competing interests exist.

**Abstract** Activating mutations in fibroblast growth factor (FGF) receptor 3 and inactivating mutations in the NPR2 guanylyl cyclase both cause severe short stature, but how these two signaling systems interact to regulate bone growth is poorly understood. Here, we show that bone elongation is increased when NPR2 cannot be dephosphorylated and thus produces more cyclic GMP. By developing an in vivo imaging system to measure cyclic GMP production in intact tibia, we show that FGF-induced dephosphorylation of NPR2 decreases its guanylyl cyclase activity in growth plate chondrocytes in living bone. The dephosphorylation requires a PPP-family phosphatase. Thus FGF signaling lowers cyclic GMP production in the growth plate, which counteracts bone elongation. These results define a new component of the signaling network by which activating mutations in the FGF receptor inhibit bone growth.
DOI: https://doi.org/10.7554/eLife.31343.001

## Introduction

Longitudinal growth of limbs and vertebrae depends on division and differentiation of chondrocytes within the cartilage growth plates located at each end of the growing bone (see Figure 2B), resulting in formation of a scaffold that is subsequently mineralized (*Kozhemyakina et al., 2015*). These processes are tightly controlled by multiple regulatory pathways. One such regulator is the natriuretic peptide-stimulated guanylyl cyclase, natriuretic peptide receptor 2 (NPR2), also known as guanylyl cyclase B, which is found in growth plate chondrocytes and promotes bone elongation (*Yasoda et al., 1998*; *Yamashita et al., 2000*; *Chusho et al., 2001*; *Tamura et al., 2004*; *Bartels et al., 2004*). Inactivating mutations in NPR2, which reduce cGMP, result in severe shortening of bones in mice and people, causing the condition acromesomelic dysplasia type Maroteaux (AMDM), in which height is reduced by ~30% (*Maroteaux et al., 1971*; *Tamura et al., 2004*; *Bartels et al., 2004*; *Khan et al., 2012*; *Geister et al., 2013*; *Nakao et al., 2015*). Conversely, activating mutations of NPR2 result in longer bones (*Miura et al., 2012*; *Hannema et al., 2013*; *Miura et al., 2014*), opposite to what is seen with the AMDM patients.

Activation of the NPR2 guanylyl cyclase requires the extracellular binding of C-type natriuretic peptide (CNP) and also the phosphorylation of multiple intracellular juxtamembrane serines and threonines (*Potter, 1998*; *Potter, 2011*) (*Figure 1A*). Previous studies have established a close

**eLife digest** Between birth and puberty, the bones of mammals grow drastically in length. This process is controlled by many proteins, and mutations affecting these proteins can cause bones to either be too long or too short. For example, mutations of a protein called the fibroblast growth factor receptor, or FGF for short, and a protein called NPR2, can cause similar forms of dwarfism – a condition characterized by short stature.

The FGF protein controls bone growth, and people with overactive receptors for FGF suffer from a form of dwarfism known as achondroplasia, while people that lack FGF receptors have longer bones. The NPR2 protein, on the other hand, produces a molecule called cGMP, which is necessary for the bones to grow. When NPR2 is blocked, less cGMP is produced, which results in shorter limbs.

Previous studies of bone cells grown in the laboratory have shown that these two proteins are linked by a chain of chemical messages. When the FGF receptor is active, phosphate molecules are removed from the NPR2 protein, which reduces the amount of GMP produced. However, until now it was not known whether this mechanism also controls growth in actual bones.

Here, Shuhaibar et al. used genetically modified mice in which the phosphate group could not be removed from their NPR2 enzyme. As a result, the bones of these mice were longer than usual. Shuhaibar et al. then developed an imaging technique to examine the region in the bone were growth happens. To see whether FGF reduces the amount of cGMP produced by NPR2 in these areas, cGMP was detected with a fluorescent sensor in order to be tracked.

In normal mice, the FGF receptor reduced the rate at which cGMP was produced, but in mice with mutated NPR2, this did not happen. When the cells could not remove the phosphates from NPR2, cGMP levels stayed high and the bones grew longer.

These findings reveal new insights into the molecular causes of dwarfism. The next step will be to identify the enzyme responsible for removing phosphate from NPR2. Blocking its activity could help to enhance bone growth. In the future, this could lead to new drug treatments for achondroplasia.
DOI: https://doi.org/10.7554/eLife.31343.002

correlation between NPR2 phosphorylation and NPR2 activity (*Abbey-Hosch et al., 2005*; *Egbert et al., 2014*; *Robinson et al., 2017*), and if these phosphorylation sites are mutated to alanine, CNP-dependent guanylyl cyclase activity is reduced to only 6% of that of the wild type protein (*Potter and Hunter, 1998*). Correspondingly, if the seven serines and threonines are mutated to the phosphomimetic amino acid glutamate (NPR2-7E), the maximal velocity of the CNP-dependent guanylyl cyclase activity is the same as that of the wild type enzyme, but does not decrease in response to stimuli that dephosphorylate and inactivate the wild type protein (*Yoder et al., 2012*; *Shuhaibar et al., 2016*; *Robinson et al., 2017*) (*Figure 1A*). Our previous analysis of mice in which both copies of *Npr2* were globally replaced with a sequence encoding NPR2-7E (*Npr2*$^{7E/7E}$) demonstrated that dephosphorylation of NPR2 is a physiological mediator of hormonal signaling in ovarian follicles (*Shuhaibar et al., 2016*). These findings led us to investigate whether the phosphorylation of NPR2 could also be a mediator of growth factor signaling in bones.

Another essential regulator of bone elongation is FGF receptor 3 (FGFR3); activating mutations of FGFR3 inhibit bone growth, causing achondroplasia, in which human height is reduced by ~25% (*Horton et al., 1978*; *Rousseau et al., 1994*; *Shiang et al., 1994*; *Naski et al., 1998*; *Wang et al., 1999*; *Lorget et al., 2012*; *Lee et al., 2017*; *Ornitz and Legeai-Mallet, 2017*). Conversely, mice and people lacking functional FGFR3 have longer bones (*Colvin et al., 1996*; *Deng et al., 1996*; *Makrythanasis et al., 2014*). Intriguingly, the greatly reduced bone length seen with activating mutations of FGFR3, resembles that seen with inactivating mutations of NPR2, and activation of NPR2 by increasing CNP opposes the decrease in bone growth caused by activating mutations of FGFR3 (*Yasoda et al., 2004*; *Yasoda et al., 2009*; *Lorget et al., 2012*; *Wendt et al., 2015*). Likewise, the increased bone length seen with mice lacking FGFR3 resembles that seen with activating mutations of NPR2 (*Miura et al., 2012*; *Hannema et al., 2013*; *Miura et al., 2014*). Furthermore, studies of fibroblasts and chondrogenic cells derived from embryonic carcinomas and chondrosarcomas have indicated that FGF signaling decreases NPR2 activity (*Chrisman and Garbers, 1999*;

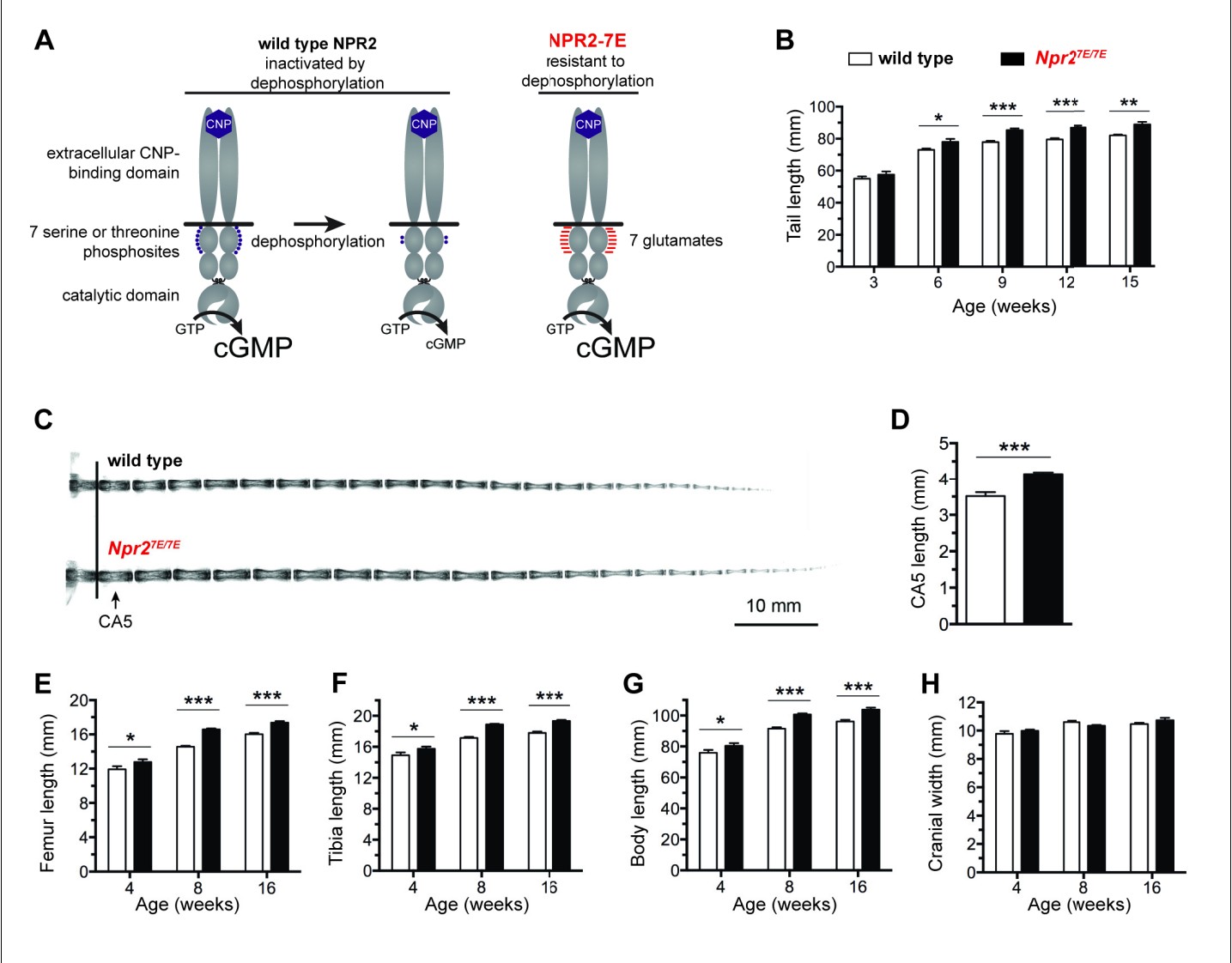

**Figure 1.** Mutation of the seven regulatory serines and threonines of NPR2 to glutamates increases bone length. (A) Regulation of NPR2 activity by the phosphorylation state of the seven juxtamembrane serines and threonines on each NPR2 monomer. Purple dots represent phosphates on these serines and threonines. Red lines represent glutamates that are substituted for the serines and threonines. (B) $Npr2^{7E/7E}$ mice have longer tails. Measurements were made from 17 to 19 live mice of each genotype. (C) X-rays of representative tails from mice euthanized at 18 weeks of age. (D) Increased CA5 vertebra length in $Npr2^{7E/7E}$ mice compared with wild type, determined from x-ray measurements of tails at 18 weeks of age (10–13 mice of each genotype). (E,F) Longer femurs (E) and tibias (F) in $Npr2^{7E/7E}$ vs wild type mice. (G) Increased body length in $Npr2^{7E/7E}$ vs wild type mice. (H) No difference in cranial width comparing $Npr2^{7E/7E}$ and wild type mice. For E–H, each bar indicates measurements from 10 to 31 mice that were euthanized at the indicated ages. All graphs show mean ±s.e.m. Data were analyzed by unpaired t-tests, with the Holm-Sidak correction for multiple comparisons where appropriate. T-tests rather than ANOVA were used because we were only interested in comparisons between genotypes at a given age, rather than comparisons across ages. *p≤0.05; **p≤0.01; ***p≤0.001.

DOI: https://doi.org/10.7554/eLife.31343.003
The following source data is available for figure 1:

**Source data 1.** Numerical data for *Figure 1B and D–H*, listing individual measurments used to calculate the means and standard errors in the corresponding graphs.
DOI: https://doi.org/10.7554/eLife.31343.004

*Ozasa et al., 2005*; *Robinson et al., 2017*), and that the inactivation is due to dephosphorylation of NPR2 (*Robinson et al., 2017*). These studies suggested to us that dephosphorylation and inactivation of NPR2 could be a mechanism by which FGF decreases elongation of bones in vivo.

Here we tested this hypothesis by investigating the growth of bones in $Npr2^{7E/7E}$ mice in which NPR2 cannot be inactivated by dephosphorylation, and found that their bones are longer than those of wild type mice. We then developed a live tissue imaging system for monitoring the guanylyl cyclase activity of NPR2 in intact growth plates from wild type and $Npr2^{7E/7E}$ mice that express a FRET sensor for cGMP. Using this system, we found that FGF-induced dephosphorylation and inactivation of NPR2 decreases cGMP production in growth plate chondrocytes, thus contributing to FGF-dependent decreases in bone growth.

## Results

### Mice in which NPR2 cannot be dephosphorylated have longer bones

$Npr2^{7E/7E}$ mice, in which NPR2 is modified to mimic a constitutively phosphorylated protein (*Shuhaibar et al., 2016*) (*Figure 1A*), have longer bones. Measurements at 8–16 weeks of age showed that the tails, femurs, tibias, and bodies of $Npr2^{7E/7E}$ mice were 8% to 14% longer than wild type (*Figure 1B,E,F,G*; *Figure 1—source data 1*). Differences in limb and body length were seen as early as 4 weeks (*Figure 1E–G*). The fifth caudal vertebra was 18% longer, as measured at 18 weeks (*Figure 1C,D*; *Figure 1—source data 1*). These results indicate that when NPR2 cannot be inactivated by dephosphorylation, endochondral bone growth is increased. The width of the cranium, which is dependent on membranous, not endochondral, bone growth was unaffected (*Figure 1H*; *Figure 1—source data 1*). The increases in body and bone length, which resulted from an inability of NPR2 to be inactivated by dephosphorylation, phenocopied those previously reported for mice in which NPR2 activity was increased by overexpression of the NPR2 agonist CNP or by an activating mutation in the NPR2 guanylyl cyclase domain (*Yasoda et al., 2004*; *Miura et al., 2012*) and in a human patient with the same activating mutation in NPR2, who was 14% taller than average at 15 years of age (*Miura et al., 2012*).

### $Npr2^{7E/7E}$ mice have longer bones due to a direct effect of the mutations on NPR2 function in the bone itself

Because NPR2 is expressed in tissues outside of the bone growth plate, including the pituitary and gastro-intestinal tract (*Tamura and Garbers, 2003*; *Sogawa et al., 2010*), we tested whether the longer bones of the $Npr2^{7E/7E}$ mice resulted from a direct action on bone rather than an indirect action on other tissues. To do this, we cultured isolated tibia from 3 to 4 day old mice, such that we could measure growth rates in the absence of extrinsic hormonal or nutritional effects.

When first dissected, $Npr2^{7E/7E}$ and wild type tibia did not differ in length (*Figure 2A–C*; *Figure 2—source data 1*). Measurements were made by adding the lengths of the proximal and distal epiphyses and the calcified ossification center (*Figure 2B*). The absence of a detectable difference in length is consistent with previous studies showing no difference in body or tibia length at birth comparing mice with inactivated NPR2 and wild type mice (*Tamura et al., 2004*). Likewise, little or no difference in body or bone length is seen at birth of humans with inactivating NPR2 mutations (*Bartels et al., 2004*). These previous studies indicate that NPR2 activity is not required for prenatal bone elongation. However, when bones of newborn mice were placed in culture, the rate of bone elongation over a 6 day period was reduced if the $Npr2$ gene was disrupted (*Tamura et al., 2004*). Correspondingly, when cultured in the presence of 1 μM CNP and measured over a period of 6 days, the rate of bone elongation was greater for $Npr2^{7E/7E}$ compared with wild type tibia (*Figure 2A,D*; *Figure 2—source data 1*). Because the growth plate is located within the epiphysis region (*Figure 2B*), we also measured the rate of elongation within this specific region. For the proximal epiphysis, there was a 14% greater elongation rate for $Npr2^{7E/7E}$ compared to wild type (*Figure 2E*; *Figure 2—source data 1*). These results show that $Npr2^{7E/7E}$ mice have longer bones due to a direct effect of the mutations on NPR2 function within the bone itself.

Interestingly, there was no effect of the $Npr2^{7E/7E}$ mutations on bone elongation in medium without added CNP (*Figure 2F,G*; *Figure 2—source data 1*). This result suggests CNP produced by the bones of 3–10 day old mice is not sufficient to activate NPR2, at least under our culture conditions.

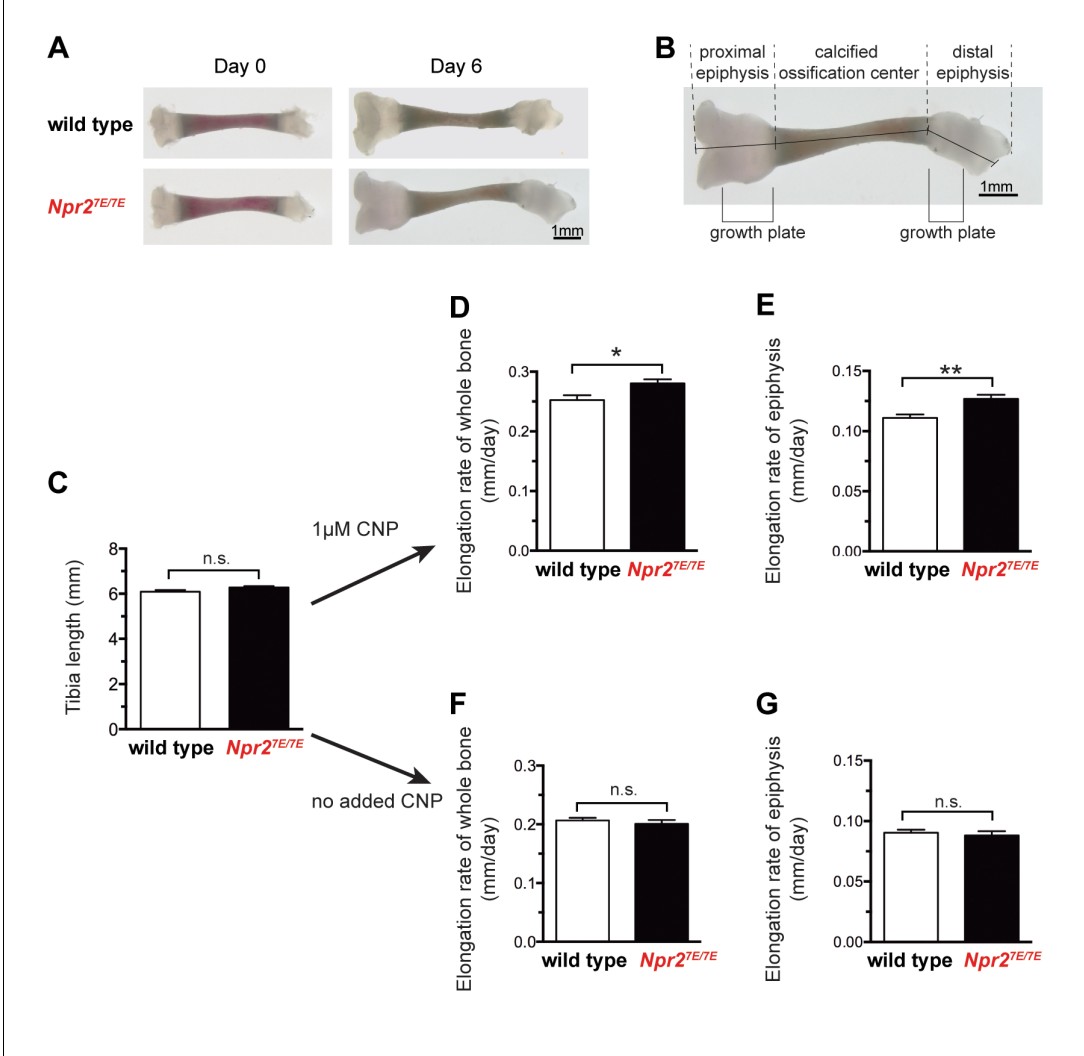

**Figure 2.** *Npr2*[7E/7E] mice have longer bones due to a direct effect on the bone. (**A**) Representative wild type and *Npr2*[7E/7E] tibia from 3 to 4 day old mice, cultured in vitro for 6 days in the presence of 1 μM CNP. (**B**) Diagram showing the methods of measurement, and the position of the growth plates. Graphs C,D, and F depict measurements of the sum of the three indicated lengths: the proximal epiphysis (left), the calcified ossification center, and the distal epiphysis (right). Graphs E and G depict measurements of the length of the proximal epiphysis only. For graphs D-G, the elongation rate was determined by subtracting the length at day 0 from the length at day 6, and dividing by 6 days. (**C**) Tibia length immediately after dissection. 44 wild type and 79 *Npr2*[7E/7E] bones were measured. (**D,E**) Rates of bone elongation in the presence of 1 μM CNP. (**D**) shows the elongation rate of the entire bone, and E shows the elongation rate of the proximal epiphysis. 21 wild type and 41 *Npr2*[7E/7E] bones were measured. (**F,G**) Rates of bone elongation in the absence of added CNP. (**F**) shows the elongation rate of the entire bone, and G shows the elongation rate of the proximal epiphysis. 23 wild type and 38 *Npr2*[7E/7E] bones were measured. Data were analyzed by unpaired t-tests. *$p \leq 0.05$; **$p \leq 0.01$.

DOI: https://doi.org/10.7554/eLife.31343.005

The following source data is available for figure 2:

**Source data 1.** Numerical data for *Figure 2C–G*, listing individual measurments used to calculate the means and standard errors in the corresponding graphs.

DOI: https://doi.org/10.7554/eLife.31343.006

## Measurements of cGMP production in living growth plates

To establish a method to investigate the signaling mechanisms underlying the increased bone growth in *Npr2*[7E/7E] mice, we developed a live imaging technique for monitoring NPR2 guanylyl

cyclase activity in intact growth plates, using mice that globally express a sensor for cGMP, cGi500 (*Russwurm et al., 2007*; *Thunemann et al., 2013*; *Shuhaibar et al., 2015*). NPR2 activity has been previously measured using homogenates of fetal tibia (*Yasoda et al., 1998*) and membrane preparations from chondrocyte cell lines (*Ozasa et al., 2005*; *Robinson et al., 2017*). The availability of mice expressing a sensor for cGMP, combined with the development of a new method to visualize chondrocytes in live and intact growth plates by confocal microscopy, allowed us to measure NPR2 activity under physiological conditions and with high spatial and temporal resolution. cGi500 is comprised of cyan fluorescent protein (CFP) and yellow fluorescent protein (YFP), linked by a cGMP-binding domain; the sensitivity of cGi500 to cGMP is based on Förster resonance energy transfer (FRET). Binding of cGMP to the linker domain causes the CFP and YFP domains to move farther apart. The proximity of CFP and YFP can be detected by imaging cells using a 440 nm laser to excite CFP; the energy emitted by CFP then excites YFP in proportion to the distance between the two fluorophores. Thus measurement of the relative intensities of the light emitted by CFP and YFP provides an indicator of the cGMP concentration. The CFP/YFP emission ratio is higher when the cGMP concentration is higher, with a half-maximal response at 500 nM cGMP, and a dynamic range of ~100 nM to ~3 µM cGMP (*Russwurm et al., 2007*).

Tibias were removed from newborn mice expressing cGi500, and placed in culture on a Millicell membrane. After 1–2 days in culture, the growth plate was exposed by cutting away the overlying tissue (*Figure 3A*), thus allowing imaging and rapid exchange of solutions. A tibia was then placed in a perfusion slide (*Figure 3B,C*) on the stage of a confocal microscope. Excitation of CFP resulted in fluorescence in all regions of the growth plate (*Figure 3D*), allowing identification of chondrocytes at different stages of development, based on the shape of the cells: round (also called resting), columnar (also called proliferating), and hypertrophic (*Kozhemyakina et al., 2015*; *Ornitz and Legeai-Mallet, 2017*). Columnar cells that are beginning to increase their volume, referred to as 'prehypertrophic', express both NPR2 and FGFR3 (*Yamashita et al., 2000*; *Chusho et al., 2001*; *de Frutos et al., 2007*; *Kozhemyakina et al., 2015*). We used this columnar/prehypertrophic region for the FRET measurements (*Figure 3E*), such that FGF effects on NPR2 activity could be investigated.

To measure the CNP-dependent guanylyl cyclase activity of NPR2, we first recorded baseline images of CFP and YFP emission in the absence of CNP. After establishing the baseline ratio, we perfused 100 nM CNP through the imaging chamber. 100 nM is close to the $EC_{50}$ for activation of NPR2 expressed in a HEK cell line (*Shuhaibar et al., 2016*). In response to CNP, the CFP emission intensity increased and the YFP emission intensity decreased (*Figure 3F*), resulting in a higher CFP/YFP ratio (*Figure 3G*) and indicating an increase in cGMP due to activation of NPR2. Thus, confocal microscopy of isolated tibia expressing cGi500 allowed us to monitor NPR2 activity in the live and intact growth plate for the first time.

The methods described here are useful for comparing rates of cGMP production by NPR2 under different experimental conditions as will be described below. However, they cannot be interpreted as a measure of cGMP levels in chondrocytes in an intact mouse, because CNP concentrations in the extracellular space around the cells of the growth plate in vivo are unknown. In our isolated tissue preparation, in which the tibia is sliced to expose the growth plate, CNP can diffuse between the extracellular space into the large volume of culture medium. Therefore the cGMP level in the absence of added CNP in the medium is not an indicator of the cGMP concentration in vivo.

## FGF reduces NPR2 guanylyl cyclase activity in the growth plate

To investigate whether FGF signaling reduces NPR2 activity within growth plate chondrocytes, we incubated cGi500-expressing tibias with or without FGF18, and imaged the fluorescence of chondrocytes in the columnar/prehypertrophic region. Among the many FGF isoforms, we used FGF18 because it is one of the two redundant FGF isoforms that function to activate FGFR3 in the growth plate (*Ornitz and Marie, 2015*; *Hung et al., 2016*). In tibias without FGF18 pretreatment, perfusion of CNP through the imaging chamber caused a large increase in the concentration of cGMP (*Figure 4A*; *Figure 4—source data 1*). However, if the tibias were pretreated with FGF18, the CNP-induced increase in cGMP was smaller (*Figure 4A,G*; *Figure 4—source data 1*). These results indicate that in chondrocytes within a living growth plate, FGF signaling reduces the activity of the NPR2 guanylyl cyclase. Similar results were obtained with FGF18 or control pretreatment times of

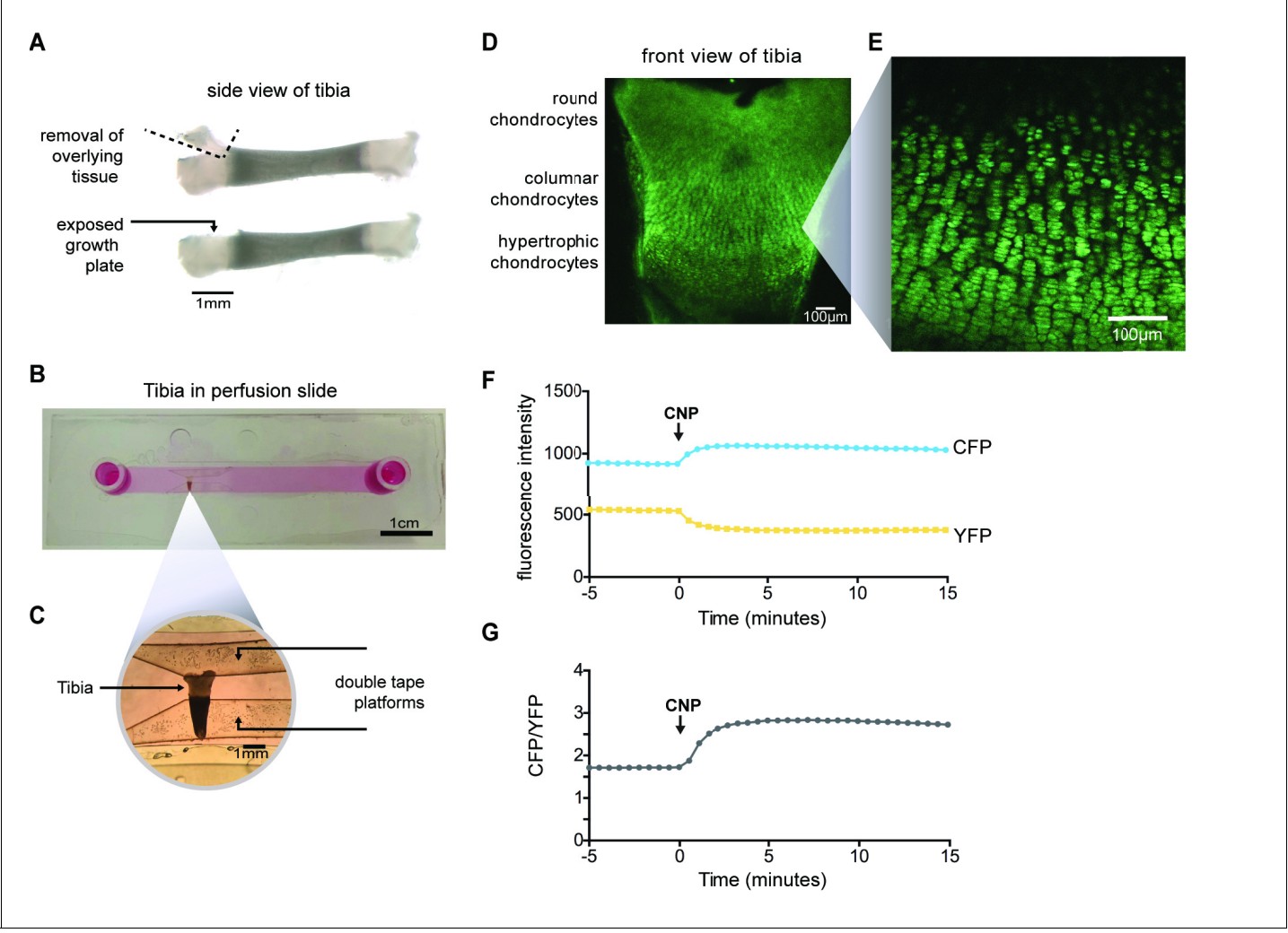

**Figure 3.** Imaging of CNP-stimulated cGMP production in chondrocytes within intact growth plates. (A) Preparation of a tibia from a newborn mouse expressing the cGi500 sensor for cGMP. (B) Tibia in a perfusion slide in which medium could be flowed across the growth plate surface while imaging. (C) Higher magnification image of the tibia in the perfusion slide. (D) Confocal image of fluorescence of the entire growth plate. (E) Higher magnification image of the columnar/prehypertrophic region that was used for measurements of cGMP (from a different tibia from that shown in D). (F, G) Time courses of CFP and YFP emission intensities and the CFP/YFP ratio, before and after perfusion of 100 nM CNP.

DOI: https://doi.org/10.7554/eLife.31343.007

80–140 min (*Figure 4A,G*), or 30–50 min (*Figure 4—figure supplement 1*; *Figure 4—source data 1*).

The baseline CFP/YFP ratio before CNP perfusion was the same with and without FGF pretreatment, suggesting that FGF had no effect on the baseline cGMP concentration (*Figure 4A*). However, since it was possible that the baseline cGMP level was lower than could be detected by the cGi500 sensor, we isolated epiphyseal regions from tibias prepared and cultured under the same conditions that were used for cGi500 imaging, and measured cGMP content using an ELISA. At this stage of limb development, almost all of the epiphysis is comprised of growth plate chondrocytes. Before CNP treatment, the cGMP content of the epiphysis was only 1–2% of that after CNP addition (*Figure 5A*; *Figure 5—source data 1*), and no significant effect of an 80 min preincubation with FGF was detected (*Figure 5B*; *Figure 5—source data 1*). These cGMP ELISA measurements also showed that in growth plates assayed after CNP addition, FGF pretreatment for 80 min decreased cGMP content to 40% of the initial level (*Figure 5A*), consistent with the cGi500 measurements.

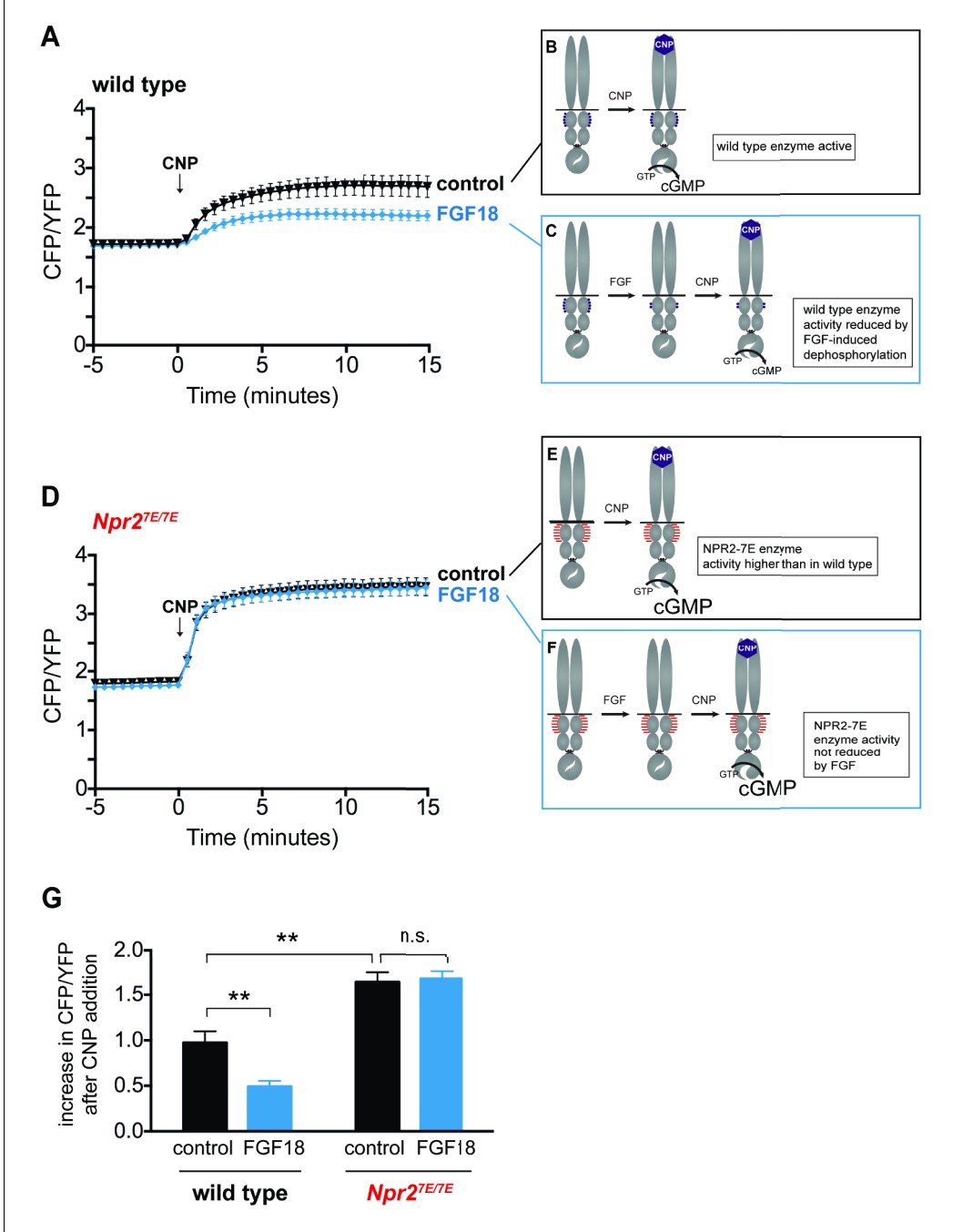

**Figure 4.** FGF reduces NPR2 guanylyl cyclase activity in growth plate chondrocytes from wild type mice, but not in those from $Npr2^{7E/7E}$ mice. Experiments were performed using tibias dissected from newborn mice (day 0–1) and used for imaging over the next 2 days. (**A**) Time course of CFP/YFP intensity ratios after 100 nM CNP was perfused across a wild type growth plate, comparing tibias that were preincubated for 80–140 min with a control solution (1 µg/ml heparin) (n = 11) or with FGF18 (0.5 µg/ml FGF18 +1 µg/ml heparin) (n = 14). CNP was present for the remainder of the recording period. An increase in the CFP/YFP ratio in response to CNP indicates cGMP production by NPR2. This and other graphs show mean ±s.e.m. (**B,C**) Schematic diagrams depicting the dephosphorylation of NPR2 in response to FGF, as demonstrated in rat chondrosarcoma cells (*Robinson et al., 2017*), and the effect on NPR2 guanylyl cyclase activity, in wild type tibia. Without FGF treatment, the seven regulatory serines and threonines on each monomer are mostly phosphorylated (depicted by five purple dots). After FGF treatment, the regulatory sites are mostly dephosphorylated (depicted by two purple dots). (**D**) Time course of CFP/YFP intensity ratios after 100 nM CNP was perfused across an $Npr2^{7E/7E}$ growth plate, comparing tibias that were preincubated for 80–140 min with a control solution (1 µg/ml heparin) (n = 10) or with FGF18 (0.5 µg/ml FGF18 +1 µg/ml heparin) (n = 11). (**E, F**) Schematic diagrams depicting NPR2 with seven glutamates on each monomer (indicated by seven red lines) and the lack of effect of FGF on NPR2 enzyme activity. (**G**) Increases in CFP/YFP ratios after CNP perfusion, comparing data from wild type tibias with and without FGF pretreatment (from **A**)

*Figure 4 continued on next page*

*Figure 4 continued*

and data from *Npr2^{7E/7E}* tibias with and without FGF pretreatment (from D). Values were determined by subtracting the mean CFP/YFP ratio 0–5 min before CNP perfusion from the mean ratio 10–15 min after CNP perfusion. Data were analyzed by unpaired t-tests, with the Holm-Sidak correction for multiple comparisons. T-tests rather than ANOVA were used because we were only interested in a subset of comparisons that tested specific hypotheses. **$p \leq 0.01$.

DOI: https://doi.org/10.7554/eLife.31343.008

The following source data and figure supplements are available for figure 4:

**Source data 1.** Numerical data for *Figures 4A,D,G* and *Figure 4—figure supplement 1A and B*, listing individual measurments used to calculate the means and standard errors in the corresponding graphs.

DOI: https://doi.org/10.7554/eLife.31343.011

**Figure supplement 1.** Inhibition of NPR2 activity by a 30–50 min preincubation with FGF.

DOI: https://doi.org/10.7554/eLife.31343.009

**Figure supplement 2.** Western blot showing that wild type and *Npr2^{7E/7E}* epiphyses contain similar amounts of NPR2 protein (25 µg total protein loaded per lane).

DOI: https://doi.org/10.7554/eLife.31343.010

## The FGF-induced decrease in guanylyl cyclase activity is caused by NPR2 dephosphorylation

Studies of the RCS chondrocyte cell line have shown that FGF signaling decreases NPR2 phosphorylation and activity, and that the decrease in activity depends on the decrease in phosphorylation (*Robinson et al., 2017*). So far, we have not succeeded in establishing methods to measure the phosphorylation state of NPR2 from growth plate tissue, and thus have not been able to determine whether the dephosphorylation seen in the chondrocyte cell line also occurs in vivo. However, we were able to use the *Npr2^{7E/7E}* mice to test whether FGF reduces the guanylyl cyclase activity of NPR2 in growth plate chondrocytes by dephosphorylating NPR2, as diagrammed in *Figure 4B,C*.

As with wild type, growth plates from *Npr2^{7E/7E}* mice showed an increase in cGMP when exposed to CNP (*Figure 4D*; *Figure 4—source data 1*). In fact, the CNP-stimulated cGMP increase was greater for *Npr2^{7E/7E}* vs wild type (*Figure 4G*; *Figure 4—source data 1*), as would be expected if NPR2 in the wild type growth plate was partially dephosphorylated as a result of signaling by endogenous hormones or growth factors (compare diagrams in *Figure 4B and E*). The higher NPR2 activity measured in the *Npr2^{7E/7E}* vs wild type growth plate was not explained by a difference in the amount of NPR2 protein, which was similar for the growth plate region of the two genotypes (*Figure 4—figure supplement 2*). Another possible explanation of the higher NPR2 activity observed for the *Npr2^{7E/7E}* genotype compared to wild type is that the 7E enzyme has a somewhat lower $K_m$ for GTP than the wildtype enzyme, although at the GTP concentrations found in most intact tissues (*Traut, 1994*), the difference in enzyme activity would be small. The 7E mutations also have no effect on the CNP concentration required to activate NPR2 to half its maximum value (*Shuhaibar et al., 2016*).

Importantly, in growth plates from *Npr2^{7E/7E}* mice, FGF pretreatment did not cause a decrease in CNP-stimulated cGMP production (*Figure 4D,G*). This lack of response to FGF contrasts with the FGF-induced decrease in NPR2 activity that is seen with wild type (*Figure 4A,G*), and indicates that because the NPR2-7E protein cannot be dephosphorylated, FGF signaling does not reduce its enzyme activity (*Figure 4F*). These results support the conclusion that FGF signaling reduces cGMP production by dephosphorylating NPR2 in the growth plate.

## A PPP-family phosphatase mediates the FGF-induced decrease in NPR2 activity

Our previous studies of NPR2 activity in ovarian follicles indicated that a PPP-family phosphatase mediates the dephosphorylation of NPR2 in response to luteinizing hormone (*Egbert et al., 2014*). To test if a PPP-family phosphatase also mediates the FGF-induced dephosphorylation of NPR2 in the growth plate, we preincubated tibias from newborn mice with the phosphatase inhibitor cantharidin. At 100 µM, cantharidin selectively inhibits the PPP family phosphatases, PPP1, PPP2 (also known as PP2A), PPP4, and PPP5 (*Swingle et al., 2007*; *Pereira et al., 2011*) but does not inhibit PPP3 (also known as PP2B or calcineurin) (*Honkanen, 1993*; *Pereira et al., 2011*) or PPM (also known as PP2C) (*Li et al., 1993*).

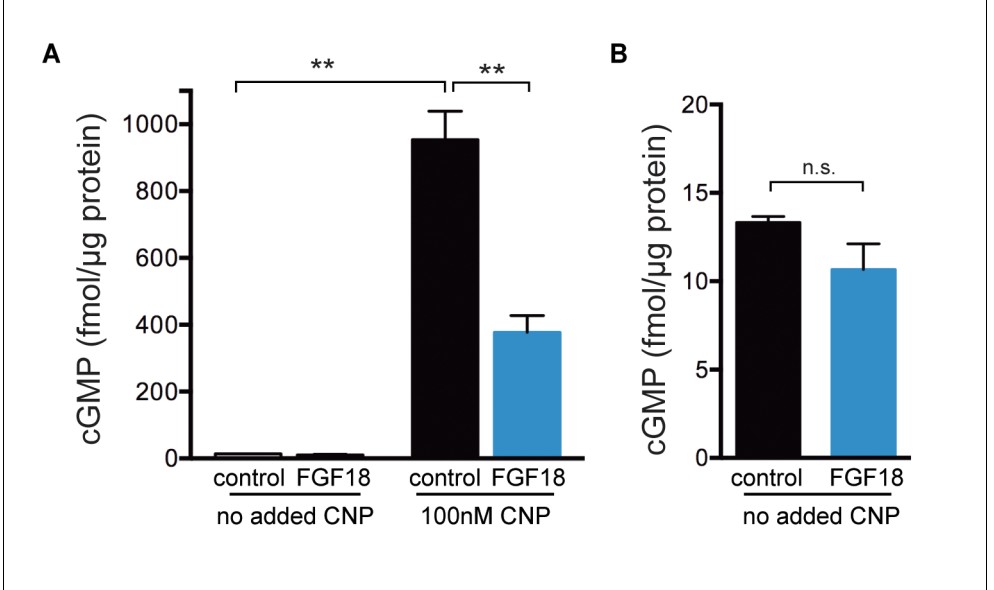

**Figure 5.** ELISA measurements of cGMP content of epiphyses of tibias from wild type newborn mice. Tibias were collected and cultured as described for the cGi500 measurements. The proximal epiphyses were then slit to expose the growth plate (see *Figure 3A*) and the slit tibias were preincubated in a well of medium containing 1 µg/ml heparin with or without FGF-18 (0.5 µg/ml) for 80 min. After the preincubation ± FGF, the epiphysis regions were cut off and prepared for cGMP ELISA measurement as described in the Materials and Methods. Where indicated, the preincubation ±FGF was followed by a 15 min incubation in the same medium, but also including 100 nM CNP, mimicking the conditions used for the cGi500 measurements. (**A**) cGMP content of the epiphyses normalized to protein content. The graph shows the mean ±s.e.m. of measurements from three separate experiments. (**B**) The values from A for the no added CNP condition are replotted with an expanded y-axis. Data were analyzed by unpaired t-tests, with the Holm-Sidak correction for multiple comparisons. **p≤0.01.

DOI: https://doi.org/10.7554/eLife.31343.012

The following source data is available for figure 5:

**Source data 1.** Numerical data for *Figure 5A,B*, listing individual measurments used to calculate the means and standard errors in the corresponding graphs.

DOI: https://doi.org/10.7554/eLife.31343.013

---

Tibias from newborn mice expressing cGi500 were pre-incubated with or without 100 µM cantharidin, then with or without FGF-18, then imaged before and after perfusion of CNP (*Figure 6*). Control tibias without cantharidin treatment showed the expected decrease in CNP-dependent cGMP production in response to FGF-18 (*Figure 6A,C*; *Figure 6—source data 1*). However, tibias that had been pre-incubated with cantharidin showed no difference in NPR2 activity comparing those that had or had not been treated with FGF-18 (*Figure 6B,C*; *Figure 6—source data 1*). These results show that like the *Npr2*[7E/7E] mutation, cantharidin inhibits the decrease in cGMP production in response to FGF. These experiments provide independent evidence that preventing the dephosphorylation of NPR2 prevents the decrease in NPR2 activity, and also indicate that a PPP family phosphatase is responsible for the dephosphorylation.

## Discussion

Our findings identify two new components of the signaling network by which FGFR3 and NPR2 signaling regulate bone growth. Firstly, our results indicate that phosphorylation of the NPR2 guanylyl cyclase promotes bone elongation; this conclusion is based on our finding that a mutation that prevents NPR2 dephosphorylation results in longer bones. Secondly, our results indicate that part of the mechanism by which FGF signaling inhibits bone elongation is by decreasing NPR2 phosphorylation and activity. The dephosphorylation and inactivation of NPR2 is mediated by a PPP-family phosphatase. However, mice lacking the FGFR3 receptor (*Colvin et al., 1996*; *Deng et al., 1996*) show a

greater increase in bone length (~7–40%) than we report here for mice with the mutation that prevents dephosphorylation of NPR2 (8–14%), consistent with previous studies identifying multiple mechanisms by which FGF signaling opposes bone elongation (*Ornitz and Legeai-Mallet, 2017*).

The working model in *Figure 7* shows our findings in the context of other knowledge about the cross-talk between FGFR3 and NPR2 signaling. An important pathway by which FGFR3 signaling decreases bone elongation is by activation of MAP kinase (see *Ornitz and Legeai-Mallet, 2017*). An important pathway by which NPR2 guanylyl cyclase activity increases bone elongation is by elevating cGMP and activating the cGMP-dependent protein kinase PRKG2 (*Pfeifer et al., 1996*; *Chikuda et al., 2004*). Previous studies have established that application of CNP to increase NPR2 guanylyl cyclase activity in chondrocytes, thus elevating cGMP and stimulating cGMP-dependent protein kinase, inhibits FGF-induced MAP kinase activation (*Yasoda et al., 2004*; *Krejci et al., 2005*; *Ozasa et al., 2005*; *Kamemura et al., 2017*), which would counteract FGF-inhibition of bone growth (solid blue line in *Figure 7*). Other mechanisms (for example, see *Kawasaki et al., 2008*) may also contribute to how PRKG2 activity increases bone growth (dotted blue line in *Figure 7*).

Our present findings identify another level of crosstalk between FGFR3 and NPR2 signaling, indicated by the orange box in *Figure 7*. By causing dephosphorylation of NPR2, FGF signaling reduces cGMP production, thus opposing bone elongation. How FGF signaling causes dephosphorylation of NPR2 remains to be determined; this could occur by inactivation of the kinase(s) that phosphorylate NPR2, and/or by activation of the phosphatase(s) that dephosphorylate it. Our results indicate that the phosphatase(s) belong to the PPP family, but do not identify the particular PPP family member (s). Activation of PPP2 is one possibility, since PPP2 is activated by FGF signaling in the RCS chondrocyte cell line (*Kolupaeva et al., 2013*), and purified PPP2 can dephosphorylate NPR2 in vitro (*Potter, 1998*).

Other groups have used fluorescence microscopy to quantify chondrocyte movement, division, and volume in live avian growth plate cartilage (*Li et al., 2015*), and to measure chondrocyte density in fixed mouse mandibular cartilage (*He et al., 2017*). However, the methods that we have developed for imaging the growth plate of mammalian bones are unique in that they allow rapid manipulation of the chemical environment surrounding the growth plate and real-time measurements of changes in signaling pathways within the intact tissue. These methods are broadly applicable to studies of signaling by other hormones and growth factors that might affect cGMP levels in growth plate chondrocytes. With mice expressing related FRET sensors for cAMP (*Calebiro et al., 2009*), these methods could also be readily applied to studies of signaling by hormones such as parathyroid hormone related protein that regulate chondrocyte and osteoblast growth and differentiation by way of cAMP (*Chagin et al., 2014*; *Kozhemyakina et al., 2015*; *Esbrit et al., 2016*).

Ongoing clinical trials have shown that in patients with achondroplasia, in which skeletal dysplasia is caused by a constitutively active form of FGFR3, stimulation of NPR2 by subcutaneous injection of a hydrolysis-resistant analog of CNP increases bone length (*Klag and Horton, 2016*). These results are consistent with the increased bone growth that results from increasing CNP in mouse models of achondroplasia (*Yasoda et al., 2004*; *Yasoda et al., 2009*; *Lorget et al., 2012*; *Wendt et al., 2015*). Studies using a mouse model overexpressing a constitutively active form of FGFR3 showed that applying CNP or a CNP analog in vivo or in vitro completely rescued the growth defect (*Yasoda et al., 2004*; *Yasoda et al., 2009*; *Wendt et al., 2015*). However, in a study involving mice in which one allele of the *Fgfr3* gene was replaced with a constitutively active form, as occurs in achondroplasia, it was found that treatment with the CNP analog increased the growth rate of cultured tibia only partially, from ~40% to~70% of wild type (*Lorget et al., 2012*). Such incomplete rescue might be expected, considering that NPR2 guanylyl cyclase activity depends not only on the presence of CNP but also on phosphorylation of multiple regulatory serines and threonines of NPR2 (*Figure 1A*). CNP levels are elevated in patients with achondroplasia, suggesting that NPR2 in their chondrocytes is resistant to CNP (*Olney et al., 2015*), which could be due to NPR2 dephosphorylation. Thus, if NPR2 phosphorylation could be increased in chondrocytes of these patients, by inhibiting the phosphatase that dephosphorylates NPR2, this could potentially enhance the therapeutic stimulation of NPR2 activity by CNP as a treatment for achondroplasia. Our findings with the phosphatase inhibitor cantharidin support the concept of using related phosphatase inhibitors (*Lu et al., 2009*; *Chung et al., 2017*) for treatment of achondroplasia.

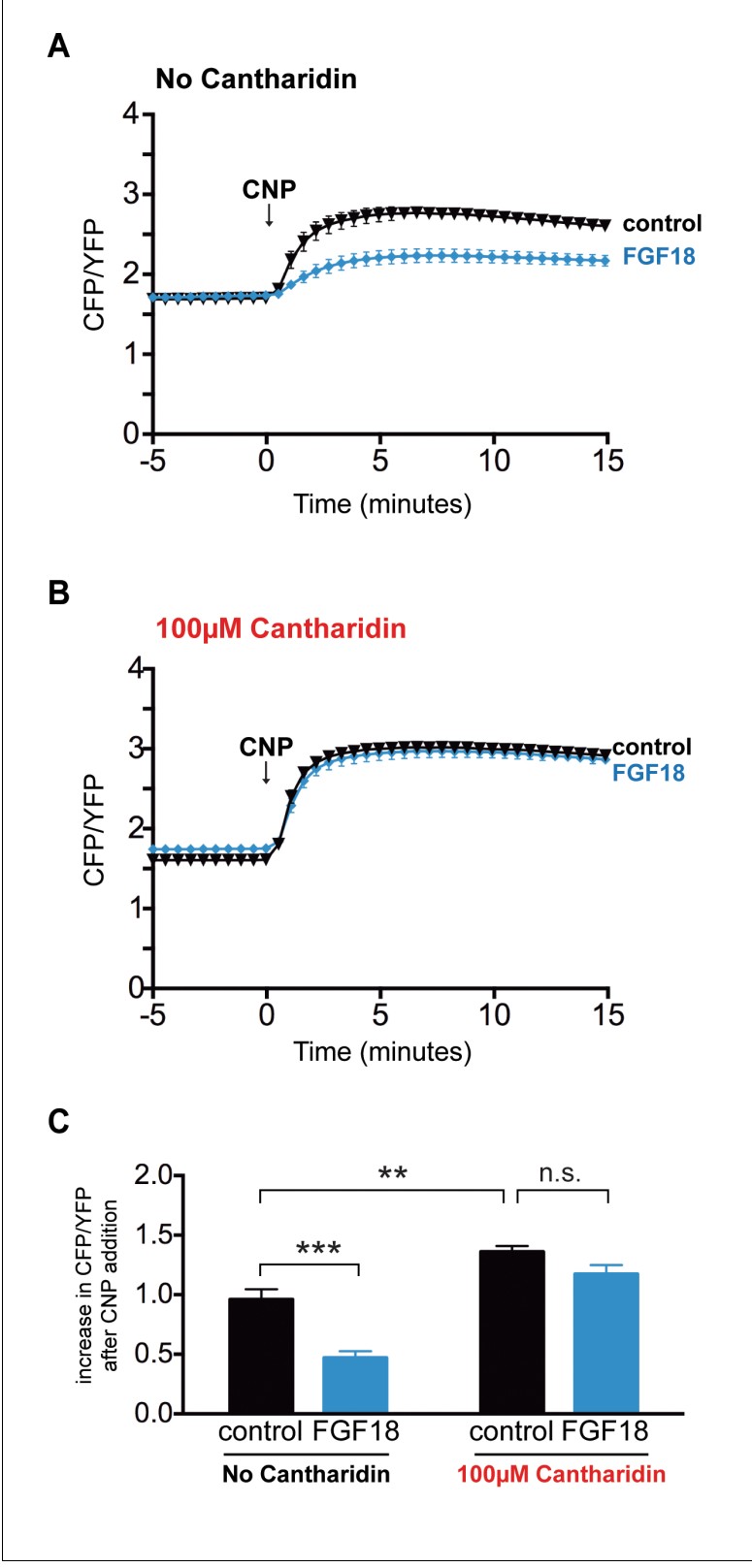

**Figure 6.** FGF does not reduce NPR2 guanylyl cyclase activity in growth plate chondrocytes of tibia that were pre-incubated with the PPP family phosphatase inhibitor cantharidin. Experiments were performed using tibias from newborn mice (day 0–1) and used for experiments over the next 1–2 days. (A,B) Time courses of CFP/YFP intensity ratios after 100 nM CNP was perfused across wild type growth plates, comparing tibias that were pre-incubated

*Figure 6 continued on next page*

*Figure 6 continued*

for 60 min with cantharidin (100 µM + 0.2% DMSO) or a control solution (0.2% DMSO), followed by a 80–120 min incubation with FGF-18 (0.5 µg/ml +1 µg/ml heparin) or a control solution (1 µg/ml heparin). (**A**) No cantharidin (0.2% DMSO only) followed by FGF-18 (n = 10) or a control solution (n = 6). (**B**) 100 µM cantharidin followed by FGF-18 (n = 10) or a control solution (n = 8). (**C**) Increases in CFP/YFP ratios after CNP perfusion, comparing data from tibias with or without cantharidin and FGF-18 pre-treatments (data from A and B). Values were determined by subtracting the mean CFP/YFP ratio 0–5 min before CNP perfusion from the mean ratio 10–15 min after CNP perfusion. Data were analyzed by unpaired t-tests, with the Holm-Sidak correction for multiple comparisons. **p≤0.01; ***p≤0.001.

DOI: https://doi.org/10.7554/eLife.31343.014

The following source data is available for figure 6:

**Source data 1.** Numerical data for *Figure 6A–C*, listing individual measurments used to calculate the means and standard errors in the corresponding graphs.

DOI: https://doi.org/10.7554/eLife.31343.015

# Materials and methods

## Key resources table

| Reagent type (species) or resource | Designation | Source or reference | Identifiers | Additional information |
| --- | --- | --- | --- | --- |
| genetic reagent (*M. musculus*) | *Npr2-7E* | PMID: 26522847 | | |
| genetic reagent (*M. musculus*) | cGi500 | PMID: 23801067 | | |
| antibody | Anti-NPR2 | *Ter-Avetisyan et al. (2014)* | | |

## Mice

Two mouse lines were used for this study: *Npr2-7E* (*Shuhaibar et al., 2016*), and cGi500 (*Thunemann et al., 2013*). The genetic background of the *Npr2-7E* line was 75% C57BL/6 and 25% 129Sv. All of the mice that were used for *Figure 1* were from the *Npr2-7E* line; about half of these mice were littermates from heterozygote x heterozygote breeding pairs. The genetic background of the cGi500 line was >90% C57BL/6. All experiments were conducted as approved by the University of Connecticut Health Center and the University of Minnesota animal care committees.

## Measurements of tail, bone, and body lengths

Tail lengths of live mice were measured at 3 week intervals. The lengths of the fifth caudal vertebrae were measured from euthanized mice, using a Faxitron cabinet x-ray system. Body lengths were measured from the tip of the nose to the base of the tail, using euthanized mice and a digital caliper. Femur and tibia lengths and cranial width were measured using excised bones and a digital caliper. For all measurements, there were approximately equal numbers of males and females of each genotype.

## Measurement of tibia growth in vitro

Tibias were dissected from 3 to 4 day old mice, and cultured as previously described (*Tamura et al., 2004*) on Millicell organotypic membranes (PICMORG50; Merck Millipore Ltd, Cork, IRL), in BGJb medium (Fitton-Jackson modification) (Life Technologies, Grand Island, NY) with 0.1% BSA, 100 units/ml each of penicillin and streptomycin, with or without 1 µM CNP (Phoenix Pharmaceuticals, Burlingame, CA). The medium was changed every 2 days. Tibias were photographed using a Leica stereoscope, for measurement of length before and after a 6 day culture period, as shown in *Figure 2B*. Measurements were done using ImageJ software (National Institutes of Health, Bethesda, MD).

## Measurement of cGMP production using cGi500

Cyclic GMP production was measured using tibias dissected from newborn mice (day 0–1) that globally expressed one or two copies of the cGi500 FRET sensor inserted into the *Rosa26* locus (*Thunemann et al., 2013*). Isolated tibias were placed on Millicell membranes, in the medium

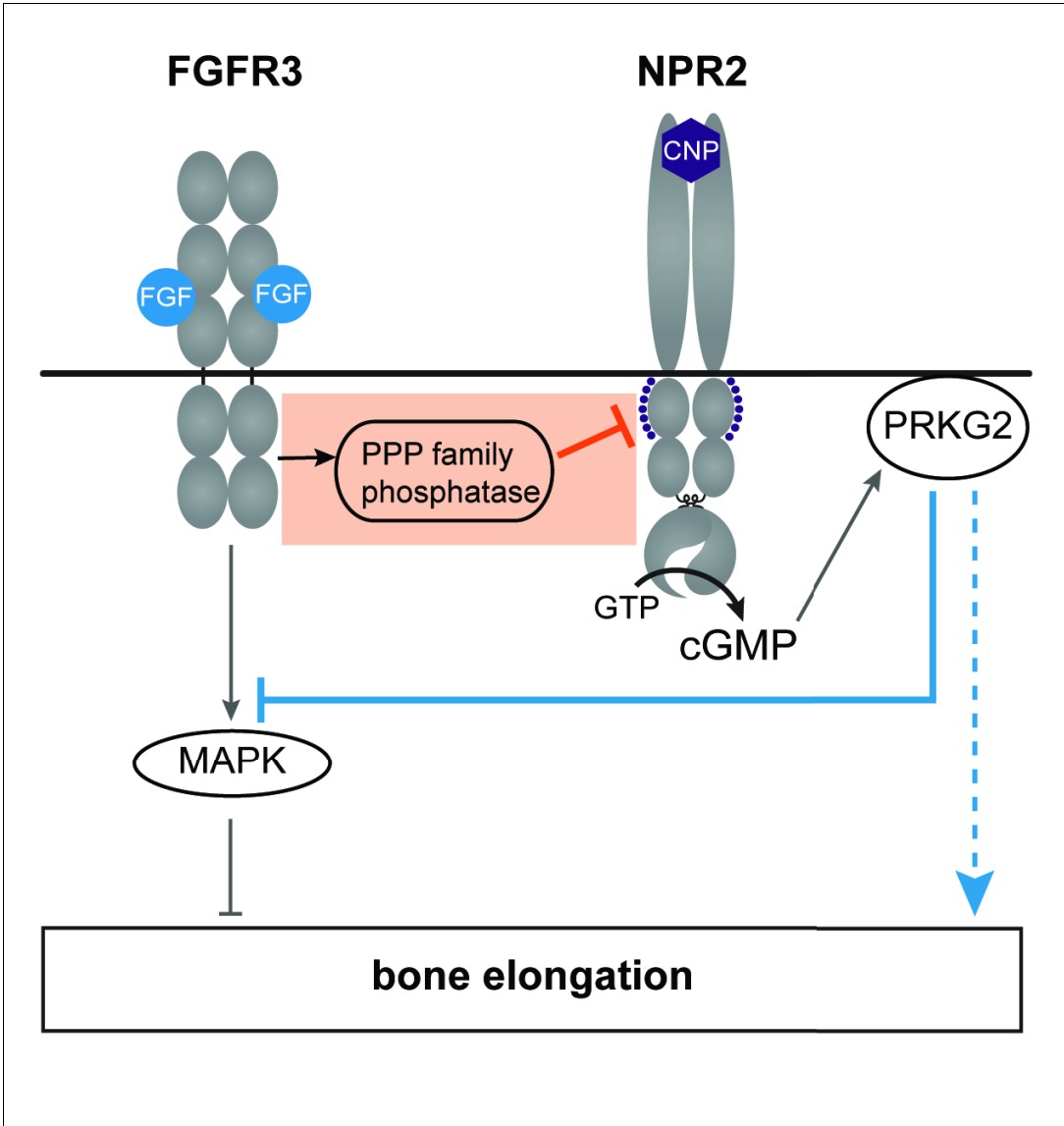

**Figure 7.** A working model of the crosstalk between NPR2 and FGFR3 signaling pathways in regulating bone elongation, emphasizing the dephosphorylation and inactivation of NPR2 in response to FGFR3 signaling (orange box). This diagram shows only some aspects of FGFR3 signaling in the growth plate; see *Karuppaiah et al. (2016)* and *Ornitz and Legeai-Mallet (2017)* for current models showing other components as well. Our findings indicate that FGF activation of its receptor FGFR3 acts by way of a PPP family phosphatase to decrease the phosphorylation and activity of the NPR2 guanylyl cyclase, thus reducing production of cGMP. Cyclic GMP increases bone growth by activating the cGMP-dependent protein kinase PRKG2. FGFR3 also acts by way of the mitogen-activated protein kinase (MAPK) to oppose bone elongation. One consequence of PRKG2 kinase activation is to inhibit MAPK (solid blue line), with the net effect of increasing bone elongation. PRKG2 may act by other pathways as well, as depicted by the dotted blue line. See main text for references.

DOI: https://doi.org/10.7554/eLife.31343.016

described above, without CNP, and were used for imaging over the next 2 days. Prior to imaging, the epiphysis region of the proximal end of the tibia was slit with a razor blade to expose the growth plate (*Figure 3A*). This was accomplished by placing the tibia, dorsal side up, in a 500 μm deep channel in a plastic slide (ibidi USA, Fitchburg, WI; cat. no. 80176, special order with no adhesive). The depth of the channel was modified from 400 to 500 μm by adding a piece of tape on each side of the channel. The epiphysis region was then slid through the edge of a razor blade on the slide surface. This procedure resulted in a 500 μm thick piece of tissue underlying the growth plate surface.

The overlying flap of tissue was trimmed away, and the tibia was placed in a 4-well plate (Nunc 176740; Thermo Scientific, Rochester, NY) containing 0.5–1 ml of medium/well at 37°C with 5% $CO_2$, on a rotating platform. Where indicated, FGF18 (PeproTech, Rocky Hill, NJ) was added at a saturating concentration (0.5 µg/ml), along with 1 µg/ml heparin (Sigma-Aldrich, H4784, St. Louis, MO). Heparin was included because it enhances FGF receptor activation (*Ornitz and Marie, 2015*). Control samples were incubated with heparin only. For phosphatase inhibitor experiments, cantharidin (Tocris BioScience, Bristol UK) was dissolved at 50 mM in DMSO and diluted in media to 100 µM. Control tibias were incubated with 0.2% DMSO.

After the indicated pre-incubations, the distal half of the tibia was cut off, and the proximal piece placed, cut edge up, in a 600 µm deep channel in a plastic slide with access wells for perfusion (ibidi USA; cat. no. 80186; special order with no adhesive) (*Figure 3B*). Silicon grease was applied around the rim of the channel. A coverslip was prepared by adhering to its surface two 200 µm thick pads that were separated by a 1–1.5 mm space. The pads were made from 2 layers of Scotch double sided tape, each 100 µm thick. The coverslip was then placed over the tibia, such that the bone spanned the two pads and the surface of the growth plate was separated by ~200 µm from the surface of the coverslip (*Figure 3C*). The coverslip was pressed down gently against the silicon grease on the perfusion slide, such that the uncut surface of the tibia was held against the perfusion slide. The slide was then inverted and filled with media, by way of ports on the slide. This resulted in an assembly in which media could be flowed under the bridge formed by the tibia resting on the tape pads. For each experiment, the separation of the growth plate surface from the coverslip was confirmed by measuring the distance between these surfaces using the confocal microscope software.

The growth plate was imaged using a Pascal confocal system (Carl Zeiss Microscopy, Thornwood, NY). Imaging was performed at ~34°C. The whole growth plate was imaged with a 10x/0.45 N.A. water immersion objective (*Figure 3D*). For FRET measurements, columnar/prehypertrophic chondrocytes were imaged using a 25x/0.8 NA water immersion objective (*Figure 3E*). To avoid evaporation during time-lapse imaging, Immersol (Carl Zeiss Microscopy) was used to form the contact with the coverslip. The pinhole was set at maximum, resulting in an optical section thickness of ~24 µm. FRET measurements were performed and analyzed as previously described (*Norris et al., 2009*; *Shuhaibar et al., 2015*). Images were collected at zoom 0.7, using 3.2 s scans at 30 s intervals, for 5 min before CNP perfusion and 15 min afterwards. Files were saved as 12-bit images. The fluorescence signal was 2–6 times the background level. Measurements were corrected for background and for spectral bleedthrough of light emitted by CFP into the YFP channel. Graphs show measurements from 490 × 490 µm regions.

## Measurement of cGMP using ELISA

To prepare samples for cGMP ELISA measurements, tibias from newborn mice were treated as described in the legend for *Figure 5*, and after the indicated incubations, 2–3 proximal epiphyses were cut off with a scissors and placed in a tube containing 200 µl of 0.1 M HCl. The samples were sonicated with a probe sonicator (model 60 Sonic Dismembrator, ThermoFisher Scientific; 3 × 10 pulses), heated for 3 min at 95°C, sonicated again (3 × 10 pulses), stored at −80°C, and analyzed for protein content (~10 µg per epiphysis; Pierce BCA assay, ThermoScientific, Rockford IL) and cGMP content (ADI-900–014 cGMP ELISA kit, Enzo Life Sciences, Farmingdale NY).

## Western blot for NPR2

To prepare samples for western blotting (*Figure 4—figure supplement 2*), epiphyses were dissected from femurs and tibias of newborn mice (day 0), and slit to expose the growth plate. Protein was solubilized by heating the tissue in 1% SDS at 95°C for 10 min. Protein content was determined by a BCA assay (~10 µg per epiphysis). 25 µg of protein was separated by SDS-PAGE electrophoresis, transferred to a nitrocellulose membrane, and probed with an antibody made in guinea pig against the extracellular domain of mouse NPR2 (*Ter-Avetisyan et al., 2014*). The antibody was a gift from Hannes Schmidt (University of Tübingen).

## Acknowledgements

We thank Liping Wang and David Rowe for help with x-ray imaging, Hannes Schmidt for his generous gift of the NPR2 antibody, and Viacheslav Nikolaev, Susan Ratzan, and Melina Schuh for helpful

discussions. We also thank the reviewers, Gail Mandel, Eric Espiner, and anonymous, for thoughtful and insightful comments. This work was supported by grants from the National Institutes of Health (R37HD014939 to LAJ, R01GM098309 to LRP, T32DK007203 to JWR, and R90DE022526 to NPS), and from the Fund for Science (to LCS, CND, LRP, and LAJ).

## Additional information

### Funding

| Funder | Grant reference number | Author |
| --- | --- | --- |
| Eunice Kennedy Shriver National Institute of Child Health and Human Development | R37HD014939 | Laurinda A Jaffe |
| National Institute of General Medical Sciences | R01GM098309 | Lincoln R Potter |
| National Institute of Diabetes and Digestive and Kidney Diseases | Postdoctoral training grant: T32DK007203 | Jerid W Robinson |
| National Institute of Dental and Craniofacial Research | Postdoctoral training grant: R90DE022526 | Ninna P Shuhaibar |
| Fund for Science | Postdoctoral scholarship (mentor) | Laurinda A Jaffe |
| Fund for Science | Postdoctoral scholarship (postdoc) | Leia C Shuhaibar |
| Fund for Science | Research grant | Caroline N Dealy Lincoln R Potter |

The funders had no role in study design, data collection and interpretation, or the decision to submit the work for publication.

### Author contributions

Leia C Shuhaibar, Laurinda A Jaffe, Conceptualization, Data curation, Formal analysis, Supervision, Funding acquisition, Validation, Investigation, Visualization, Methodology, Writing—original draft, Project administration, Writing—review and editing; Jerid W Robinson, Conceptualization, Data curation, Formal analysis, Supervision, Funding acquisition, Validation, Investigation, Visualization, Methodology, Project administration, Writing—review and editing; Giulia Vigone, Conceptualization, Data curation, Formal analysis, Funding acquisition, Investigation, Methodology, Writing—review and editing; Ninna P Shuhaibar, Funding acquisition, Investigation, Methodology, Writing—review and editing; Jeremy R Egbert, Tracy F Uliasz, Data curation, Formal analysis, Validation, Investigation, Methodology, Writing—review and editing; Valentina Baena, Formal analysis, Investigation, Methodology, Writing—review and editing; Deborah Kaback, Resources, Data curation, Formal analysis, Validation, Investigation, Methodology, Writing—review and editing; Siu-Pok Yee, Resources, Formal analysis, Investigation, Methodology, Writing—review and editing; Robert Feil, Resources, Methodology, Writing—review and editing; Melanie C Fisher, Resources, Investigation, Methodology, Writing—review and editing; Caroline N Dealy, Resources, Funding acquisition, Methodology, Writing—review and editing; Lincoln R Potter, Conceptualization, Resources, Formal analysis, Funding acquisition, Methodology, Project administration, Writing—review and editing

### Author ORCIDs

Leia C Shuhaibar http://orcid.org/0000-0003-4152-6263
Laurinda A Jaffe http://orcid.org/0000-0003-2636-5721

### Ethics

Animal experimentation: All experiments were conducted as approved by the animal care committees of the University of Connecticut Health Center (101395-0519) and the University of Minnesota (1507-32769A).

Decision letter and Author response
Decision letter https://doi.org/10.7554/eLife.31343.019
Author response https://doi.org/10.7554/eLife.31343.020

## Additional files

Supplementary files
• Transparent reporting form
DOI: https://doi.org/10.7554/eLife.31343.017

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
