## [Decision Letter]

Thank you for submitting your article "Dephosphorylation of the NPR2 guanylyl cyclase contributes to inhibition of bone growth by fibroblast growth factor" for consideration by *eLife*. Your article has been favorably evaluated by Fiona Watt (Senior Editor) and three reviewers, one of whom, Gail Mandel (Reviewer #1), is a member of our Board of Reviewing Editors. The following individual involved in review of your submission has agreed to reveal their identity: Eric Espiner (Reviewer #2).

The reviewers have discussed the reviews with one another and the Reviewing Editor has drafted this decision to help you prepare a revised submission.

Summary:

This work illustrates the use of a sensor for cGMP production, in a new intact preparation of mouse bone growth, to show in real time measurements that the NPR2 guanylyl cyclase is part of the FGF pathway for impairing bone growth. Prior studies of NPR2 GC activity have been confined to extracts from cell lines, and while mutations in both proteins cause forms of human dwarfism, whether the two proteins represent parallel or interdependent pathways is not known.

Essential revisions:

The reviews are explicit for concerns raised by the reviewers, and all three full reviews are attached for your consideration. While we request a point by point response to all of the concerns, in particular, in a revised manuscript, we expect a substantial revision. The major points that are a concern are:

1) Although perhaps not the point of the figure, please add data or provide explanation of tibial growth in the absence of CNP.

2) Please address whether there are any changes in cGMP levels in WT tibia in the absence of CNP. While we realize that your study measures the production of cGMP, and not basal levels, it might strengthen the real-time measurements if you could provide more information on the basal levels.

3) Related to this point, if feasible, an independent experiment supporting the idea that preventing dephosphorylation prevents the FGF effect in the absence of CNP, such as use of a phosphatase inhibitor, would strengthen the study.

Reviewer #1:

Previous studies have shown that the FGF3 receptor (FGFR3) pathway and, independently, the natriuretic peptide receptor 2, NPR2, are involved in bone growth. Mutations in both proteins cause forms of human dwarfism. However, whether the two proteins represent parallel or interdependent pathways is not known. Here, authors test whether NPR2 is in the FGF3R pathway in the physiological context of mouse bone growth. Using a new and elegant ex vivo preparation of intact tibia from mice expressing a form of NPR2 that cannot be dephosphorylated, authors show that bones are longer in these mice compared to WT mice. Then, by monitoring NPR2 guanylyl cyclase activity, in the intact growth plates, they show that FGF-dependent decreases in bone growth involve FGF-induced dephosphorylation of NPR2, with resulting decreases in cGMP in the growth plate chondrocytes. Thus, part of the mechanism by which FGF signaling inhibits bone elongation is through decreasing NPR2 activity. Of note, the intact system, in combination with the sensor, allow for rapid manipulation of the growth milieu as well as quantitating, in real time, changes in the signaling pathway. Its use here should elevate interest in using it for cGMP measurements in many other physiological contexts in addition to bone growth. Thus, the work will appeal to a wide range of developmental scientists, outside of those having a particular interest in bone growth.

I have one major concern that authors should consider addressing in a revised manuscript. While the authors point out that mutations in NPR2 result in a human form of dwarfism, the effects on bone length that they see in the mouse with the NPR2 phospho mimic mutant (~8-14%) are perhaps smaller than one might expect if the mouse model recapitulates the human condition. Authors should clarify in the Discussion their interpretation of the mouse bone length phenotype, observed in this study, with regard to the human phenotype resulting from the NPR2 mutation. They do have a nice discussion relating CNP levels and rescue of FGFR3 achondroplasia, but I could not easily relate their mouse results with the patient NPR2 dwarfism phenotype (the acromesomelic dysplasia Maroteaux type dwarfism) mentioned in the Introduction. Otherwise, with the exception of a couple of things noted below, the paper is well written for a broad audience and the data is very clearly presented.

The second sentence in the Abstract is very confusing with "NPR2 cannot be dephosphorylated" and then "is opposed when NPR2 is dephosphorylated. In Results, the use of the word intrinsic to describe the muscle effects is a bit confusing/awkward. For example, the title in the Results section: *Npr2^7E/7E^* mice have longer bones due to an intrinsic effect on bone, doesn't seem quite right. I know what they mean, but not sure this is the best way to present it. Might "direct effect of phosphorylation" be better?

Figure 1. Interesting that the tail length seems to dramatically increase between 3 and 6 weeks, does this mean there is normally a post-natal window during which cGMP levels matter? Authors indicate that NPR2 activity is not required for prenatal bone elongation, but do they have any idea what is happening during that first postnatal period?

Figure 4: What does the blue dotted arrow signify? This should be defined in the legend along with MAPK and PRKG2.

Reviewer #2:

Having shown that dephosphorylation of nprb by FGFR3 reversibly impairs GC-B activity – but not a phosphomimetic mutant – in rat chondrosarcoma cells (recent report in Cell Signaling), the current study reports that this molecular action of FGF contributes to impaired long bone growth in vivo in mice. The evidence for this is based on i) mitigated inhibition of FGF induced long bone growth in mice with genetic modifications in the CNP receptor preventing dephosphorylation and ii) novel imaging techniques, used ex vivo in intact tissues, showing concordant changes in cGMP within growth plate chondrocytes depending on the phosphorylated state of nprb induced by FGF activity.

General Comments. While not surprising in view of their recent report in Cell Signaling, the findings are important new observations that make significant contributions to our understanding of the molecular events underpinning the interplay of two important pathways regulating mammalian long bone growth. The hypotheses are clearly stated and logically pursued in the experimental approach. The methods, genetic models and analyses all appear to be valid, as are the interpretation of the findings.

Specific comments.

1) In the Introduction, the authors appear to extrapolate findings from NO induced cGMP effects on bone density (JBMR 2017) to similar effects induced by CNP activation of cGMP (Calcif Tiss Int 2012). However the report of Kondo et al. relates to increased bone turnover-not increased bone density. To date, there is no convincing evidence that CNP promotes bone density per se.

2) Introduction, third paragraph. The authors refer to mice lacking FGFR3 having longer bones: they should also cite a similar phenotype in humans with FGFR3 LOF (Makrythanasis Human Mut 2014).

3) Figure 3 legend should specify the age of the mice tibia (presumably new born, 0-2 days as stated in the Materials and methods).

Reviewer #3:

In humans and mice, the short bone phenotype caused by activating mutations in FGFR3 resembles that seen with loss-of-function mutations in NPR2 (or its ligand CNP), and pharmacological doses of hydrolysis-resistant CNP in humans or over-expression of CNP in transgenic mice largely corrects the mutant FGFR3 phenotype. Activation of NPR2 requires both receptor phosphorylation (on seven S/T residues) and binding of CNP; FGF treatment of chondroblastic cells causes NPR2 de-phosphorylation and reversible inhibition of CNP-induced NPR2 activation (as described in an accompanying manuscript by the authors). In the present study, the authors want to test the hypothesis that FGF signaling decreases NPR2 activity by dephosphorylating the receptor in vivo, using existing mice with ubiquitous expression of a cGMP-sensitive FRET sensor and mice carrying a phospho-mimetic mutant NPR2(7E).

Consistent with previous data in mice and humans with activating NPR2 mutations, the authors find that the NPR3(7E) mice have longer bones than wild type mice, and tibiae from mutant mice cultured in the presence of CNP grow faster than tibiae from wild type mice. These data suggest either increased basal guanylate cyclase activity of the mutant receptor, or enhanced sensitivity to CNP. Unfortunately, tibial growth in the absence of CNP was not evaluated.

The authors demonstrate the feasibility of measuring CNP-induced cGMP generation in intact growth plates in bones isolated from cGMP sensor mice, thereby expanding the number of live tissues in which cGMP generation can be monitored in real time. It appears that the basal CFP/YFP emission ratio in the absence of CNP varies between growth plate preparations (e.g., Figure 2 and Figure 3), but may not accurately reflect basal cGMP tissue concentrations – this should be discussed. CNP treatment results in a robust FRET change that is larger in mutant compared to wild type growth plates, and pre-treatment with FGF reduces cGMP accumulation in wild type, but not mutant growth plates. The authors conclude that "FGF signaling reduces cGMP production by dephosphorylating NPR2 in the growth plate." However, there is no data presented (or cited) to correlate NPR2 phosphorylation with GC activity. The 7E substitutions may or may not alter receptor conformation and function the same way phosphorylation does (this should be discussed in light of data shown in the accompanying manuscript).

No difference in basal cGMP concentrations could be detected between mutant and wild type growth plates, but the assay may not be suitable to detect small changes in basal cGMP concentrations. Therefore, one cannot conclude that NPR2(7E) chondrocytes in vivo have higher cGMP concentrations compared to wild type chondrocytes (as stated at the end of the Results) – but one can speculate that this may be the case given the bone overgrowth phenotype of NPR2(7E) mice. Does treatment with FGF (or phosphatase inhibitors) decrease the CFP/YFP ratio in wild type growth plates in the absence of CNP?

Overall, experiments were carefully executed and clearly described, but not all conclusions are supported by the data shown. For some of the conclusions, the biochemical properties of the mutant receptor need to be discussed and contrasted to the wild type receptor (e.g., basal and maximal activity, Ka for CNP, Km for GTP, mostly published). The most novel aspect of the study is the measurement of CNP-induced cGMP concentrations in growth plates, opening the door to assessing the effects of other hormones on cyclic nucleotides. However, the data presented are sparse – there are only two figures and the illustration of the FRET method. Figure 1 and Figure 3 could be even shorter (e.g. lengths of tail and body, vertebrae, femurs and tibiae do not need to be shown in separate graphs), and the explanatory schemata of NPR2 phosphorylation/ activity are redundant between Figure 1, Figure 3, and 4 (Figure 4 could be eliminated).

---

## [Author Response]

Essential revisions:The reviews are explicit for concerns raised by the reviewers, and all three full reviews are attached for your consideration. While we request a point by point response to all of the concerns, in particular, in a revised manuscript, we expect a substantial revision. The major points that are a concern are:1) Although perhaps not the point of the figure, please add data or provide explanation of tibial growth in the absence of CNP.

We expanded the data on *Npr2^7E/7E^*and wildtype tibia growth in culture to include results of culturing tibias without CNP. The results without CNP are presented in Figure 2, and show that in the absence of CNP, there is no difference in the rate of bone elongation comparing wild type and *Npr2^7E/7E^*. We also improved the method of analysis used for Figure 2, by including measurements of the length of the epiphysis region (which is comprised mostly of the growth plate) (see Figure 2), thus focusing on the specific region of the bone (the growth plate) that is affected by the mutation. Compared to wild type, *Npr2^7E/7E^*tibias cultured in the presence of CNP showed a 14% higher rate of elongation of the epiphysis (Figure 2).

2) Please address whether there are any changes in cGMP levels in WT tibia in the absence of CNP. While we realize that your study measures the production of cGMP, and not basal levels, it might strengthen the real-time measurements if you could provide more information on the basal levels.

We addressed the question of basal cGMP by adding ELISA measurements of cGMP in epiphyses of newborn mice (new Figure 5). We used an ELISA method because as noted by the reviewer, it is possible that basal levels are too low to be measured using the cGi500 FRET sensor. In the absence of CNP, FGF does not significantly decrease basal cGMP. Furthermore, since these basal values are only 1-2% ofthose measured by ELISA in the presence of CNP (Figure 5), they do not affect our use of cGi500 to measure CNP-dependent cGMP production with and without FGF (Figure 4).

3) Related to this point, if feasible, an independent experiment supporting the idea that preventing dephosphorylation prevents the FGF effect in the absence of CNP, such as use of a phosphatase inhibitor, would strengthen the study.

We added experiments to test the effect of the PPP family phosphatase inhibitor cantharidin on the FGF-induced decrease in cGMP production (new Figure 6). Like the *Npr2^7E/7E^*mutation, cantharidin inhibits the decrease in CNP-dependent cGMP production in response to FGF. These experiments provide independent evidence that preventing dephosphorylation prevents the decrease in NPR2 activity, and also indicate that a PPP family phosphatase is responsible for the dephosphorylation. This finding supports the concept of using related phosphatase inhibitors for treatment of achondroplasia.

Reviewer #1:[…] I have one major concern that authors should consider addressing in a revised manuscript. While the authors point out that mutations in NPR2 result in a human form of dwarfism, the effects on bone length that they see in the mouse with the NPR2 phospho mimic mutant (~8-14%) are perhaps smaller than one might expect if the mouse model recapitulates the human condition. Authors should clarify in the Discussion their interpretation of the mouse bone length phenotype, observed in this study, with regard to the human phenotype resulting from the NPR2 mutation. They do have a nice discussion relating CNP levels and rescue of FGFR3 achondroplasia, but I could not easily relate their mouse results with the patient NPR2 dwarfism phenotype (the acromesomelic dysplasia Maroteaux type dwarfism) mentioned in the Introduction. Otherwise, with the exception of a couple of things noted below, the paper is well written for a broad audience and the data is very clearly presented.

Since the NPR2-7E mutation has not been described in humans, the closest comparison that we can make is to patients with activating mutations in the NPR2 catalytic domain that constitutively activates the enzyme and thus resembles the increased activity seen with the *Npr2^7E/7E^* genotype. These human mutations result in overly long bones. For example one such patient showed a 14% increase in height (at 15 years of age) compared to the mean (Miura et al., 2012), an effect similar in magnitude to what we see with *Npr2^7E/7E^*. In the revised manuscript, we mention this comparison in discussing our results. In contrast, patients with acromesomelic dysplasia type Maroteaux (ADMD) show an ~ 30% *decrease* in height (Khan et al., 2012), rather than an increase (due to lack of functional NPR2). We revised the Introduction to clarify that the mutation we describe is due to an opposite change in NPR2 function compared to that occurring in ADMD. Like the human NPR2 mutation that increases NPR2 activity and bone growth, the *Npr2^7E/7E^* genotype has a relatively modest effect on bone length. However, mechanistic understanding of this mutation could contribute to identification of strategies to oppose other conditions that result in severe decreases in height.

The second sentence in the Abstract is very confusing with "NPR2 cannot be dephosphorylated" and then "is opposed when NPR2 is dephosphorylated.

We reworded the Abstract to avoid this confusion.

In Results, the use of the word intrinsic to describe the muscle effects is a bit confusing/awkward. For example, the title in the Results section: Npr2^7E/7E^ mice have longer bones due to an intrinsic effect on bone, doesn't seem quite right. I know what they mean, but not sure this is the best way to present it. Might "direct effect of phosphorylation" be better?

We revised the wording as suggested.

Figure 1. Interesting that the tail length seems to dramatically increase between 3 and 6 weeks, does this mean there is normally a post-natal window during which cGMP levels matter? Authors indicate that NPR2 activity is not required for prenatal bone elongation, but do they have any idea what is happening during that first postnatal period?

NPR2 is expressed in the growth plate even before birth (Yamashita et al., 2000), so it might seem surprising that no difference in bone length is seen at birth. One possible explanation is that the NPR2 agonist, CNP, might not be expressed until after birth. Possibly growth becomes NPR2 dependent in parallel with the development of CNP expression within the first few weeks after birth. To our knowledge, the developmental time course of CNP expression in the growth plate has not been examined.

Figure 4: What does the blue dotted arrow signify? This should be defined in the legend along with MAPK and PRKG2.

This information was added to the figure legend (now Figure 7).

Reviewer #2:[…] Specific comments.1) In the Introduction, the authors appear to extrapolate findings from NO induced cGMP effects on bone density (JBMR 2017) to similar effects induced by CNP activation of cGMP (Calcif Tiss Int 2012). However the report of Kondo et al. relates to increased bone turnover-not increased bone density. To date, there is no convincing evidence that CNP promotes bone density per se.

We omitted the sentence and references about bone density.

2) Introduction, third paragraph. The authors refer to mice lacking FGFR3 having longer bones: they should also cite a similar phenotype in humans with FGFR3 LOF (Makrythanasis Human Mut 2014).

Thank you for pointing out this reference, which we added to the Introduction.

3) Figure 3 legend should specify the age of the mice tibia (presumably new born, 0-2 days as stated in the Materials and methods).

We added this information to the legend of the current Figure 4.

Reviewer #3:[…] Consistent with previous data in mice and humans with activating NPR2 mutations, the authors find that the NPR3(7E) mice have longer bones than wild type mice, and tibiae from mutant mice cultured in the presence of CNP grow faster than tibiae from wild type mice. These data suggest either increased basal guanylate cyclase activity of the mutant receptor, or enhanced sensitivity to CNP. Unfortunately, tibial growth in the absence of CNP was not evaluated.

We added data addressing this question (Figure 2). This figure shows that in the absence of CNP, the rate of bone elongation for tibias in culture is the same for wild type and *Npr2^7E/7E^*.

The authors demonstrate the feasibility of measuring CNP-induced cGMP generation in intact growth plates in bones isolated from cGMP sensor mice, thereby expanding the number of live tissues in which cGMP generation can be monitored in real time. It appears that the basal CFP/YFP emission ratio in the absence of CNP varies between growth plate preparations (e.g., Figure 2 and Figure 3), but may not accurately reflect basal cGMP tissue concentrations – this should be discussed.

Thank you for pointing out that in the example shown in Figure 2 (now Figure 3) the CFP/YFP ratio before CNP perfusion (1.5) was somewhat lower than the mean ratio (1.7) shown in Figure 3 (now Figure 4). We replaced this example with a more representative record. In the revised manuscript, we also included an additional set of recordings (Figure 6) that confirm that the mean CFP/YFP ratio before CNP perfusion is consistently 1.7.

CNP treatment results in a robust FRET change that is larger in mutant compared to wild type growth plates, and pre-treatment with FGF reduces cGMP accumulation in wild type, but not mutant growth plates. The authors conclude that "FGF signaling reduces cGMP production by dephosphorylating NPR2 in the growth plate." However, there is no data presented (or cited) to correlate NPR2 phosphorylation with GC activity.

In the revised manuscript, we added a sentence to the Introduction to refer to previous papers that correlate NPR2 phosphorylation and activity. We also added a sentence to the Results to clarify that we have not yet succeeded in establishing methods to measure the phosphorylation state of NPR2 from growth plate tissue, and thus have not been able to determine whether the dephosphorylation seen in the chondrocyte cell line (Robinson et al., 2017) also occurs in vivo.

The 7E substitutions may or may not alter receptor conformation and function the same way phosphorylation does (this should be discussed in light of data shown in the accompanying manuscript).

To address the question of whether our results could be explained by an effect of the 7E mutation on an aspect of NPR2 function unrelated to phosphorylation state, we added experiments with a phosphatase inhibitor (Figure 6) that provide further evidence for our interpretation. We also added 2 sentences to the Results to clarify that the K_m_ for GTP is only slightly different for NPR2-7E and wildtype proteins, and that the EC_50_ for CNP is the same.

No difference in basal cGMP concentrations could be detected between mutant and wild type growth plates, but the assay may not be suitable to detect small changes in basal cGMP concentrations. Therefore, one cannot conclude that NPR2(7E) chondrocytes in vivo have higher cGMP concentrations compared to wild type chondrocytes (as stated at the end of the Results) – but one can speculate that this may be the case given the bone overgrowth phenotype of NPR2(7E) mice.

We omitted the statement that chondrocyte cGMP concentrations are higher in *Npr2^7E/7E^*mice compared to wild type, and clarified that our measurements provide information about rates of cGMP production for the 2 genotypes, but not about cGMP concentrations in vivo.

Does treatment with FGF (or phosphatase inhibitors) decrease the CFP/YFP ratio in wild type growth plates in the absence of CNP?

Pre-incubation with FGF does not decrease the CFP/YFP ratio measured in the absence of CNP (Figure 4, Figure 6), or the cGMP content of epiphyses incubated in the absence of CNP (Figure 5). There is also little or no effect of the phosphatase inhibitor cantharidin on the CFP/YFP ratio measured in the absence of CNP (compare baselines in Figure 6).

Overall, experiments were carefully executed and clearly described, but not all conclusions are supported by the data shown. For some of the conclusions, the biochemical properties of the mutant receptor need to be discussed and contrasted to the wild type receptor (e.g., basal and maximal activity, Ka for CNP, Km for GTP, mostly published).

This information was added to the text. In relation to this issue, we also added a western blot showing that the level of NPR2 protein expression, normalized to total protein, is the same for wild type and mutant NPR2 (Figure 4—figure supplement 2). These additions clarify that the K_m_, EC_50_, and protein expression levels of NPR2-7E are similar to the values obtained for the wild type receptor.

The most novel aspect of the study is the measurement of CNP-induced cGMP concentrations in growth plates, opening the door to assessing the effects of other hormones on cyclic nucleotides. However, the data presented are sparse – there are only two figures and the illustration of the FRET method.

The revised manuscript includes new data, 3 additional figures, and an additional supplementary figure.

Figure 1 and Figure 3 could be even shorter (e.g. lengths of tail and body, vertebrae, femurs and tibiae do not need to be shown in separate graphs).

We prefer to keep the separate graphs since we think that they effectively present the experimental results.

The explanatory schemata of NPR2 phosphorylation/ activity are redundant between Figure 1, Figure 3, and 4 (Figure 4 could be eliminated).

An improved version of Figure 4 (now Figure 7) is included in the revised manuscript. The revised model highlights the new information contributed by this study, and puts our findings in the context of other knowledge about this signaling pathway.